# The Mediating Effects of Protective Behavioral Strategies on the Relationship between Addiction-Prone Personality Traits and Alcohol-Related Problems among Emerging Adults

**DOI:** 10.3390/ijerph18041814

**Published:** 2021-02-12

**Authors:** Raquel Nogueira-Arjona, Kara Thompson, Athena Milios, Alyssa Maloney, Terry Krupa, Keith S. Dobson, Shu-Ping Chen, Sherry H. Stewart

**Affiliations:** 1Department of Psychology and Neuroscience, Dalhousie University, Halifax, NS B3H 4R2, Canada; sstewart@dal.ca; 2Department of Psychology, Saint Francis Xavier University, Antigonish, NS B2G 2W5, Canada; kdthomps@stfx.ca (K.T.); Alyssa.Maloney@dal.ca (A.M.); 3Department of Psychiatry, Dalhousie University, Halifax, NS B3H 2E2, Canada; at254917@dal.ca; 4School of Rehabilitation Therapy, Queen’s University, Kingston, ON K7L 3N6, Canada; terry.krupa@queensu.ca; 5Department of Psychology, University of Calgary, Calgary, AB T2N 1N4, Canada; ksdobson@ucalgary.ca; 6Department of Occupational Therapy, University of Alberta, Edmonton, AB T6G 2G4, Canada; shuping2@ualberta.ca

**Keywords:** alcohol-related problems, protective behavioral strategies, personality, risk factors, harm-reduction

## Abstract

Alcohol consumption and associated harms are an issue among emerging adults, and protective behavioral strategies (PBS) are actions with potential to minimize these harms. We conducted two studies aimed at determining whether the associations of at-risk personality traits (sensation-seeking [SS], impulsivity [IMP], hopelessness [HOP], and anxiety-sensitivity [AS]) with increased problematic alcohol use could be explained through these variables’ associations with decreased PBS use. We tested two mediation models in which the relationship between at-risk personality traits and increased problematic alcohol use outcomes (Study 1: Alcohol volume; Study 2: Heavy episodic drinking and alcohol-related harms) was partially mediated through decreased PBS use. Two samples of college students participated (N_1_ = 922, Mage_1_ = 20.11, 70.3% female; N_2_ = 1625, Mage_2_ = 18.78, 70.3% female). Results partially supported our hypotheses, providing new data on a mechanism that helps to explain the relationships between certain at-risk personality traits and problematic alcohol use, as these personalities are less likely to use PBS. In contrast, results showed that AS was positively related to alcohol-related harms and positively related to PBS, with the mediational path through PBS use being protective against problematic alcohol use. This pattern suggests that there are other factors/mediators working against the protective PBS pathway such that, overall, AS still presents risks for alcohol-related harms.

## 1. Introduction

Alcohol is the most frequently used drug among Canadian emerging adults [1]. According to the Canadian Tobacco, Alcohol and Drugs Survey ([2], p. 7), most 20- to 24-year-olds (83.5%) report having consumed alcohol in the past 12 months. Further, almost half (40.5%) reported heavy episodic drinking, defined as consuming 5 or more drinks (males) or 4 or more drinks (females) on a single drinking occasion at least once a month in the past year [2]. Both these rates are significantly higher than in other age groups. Reports from a large Canada-wide study of college students revealed severe harms associated with alcohol use: 35.0% did something they later regretted, 27.0% reported blackouts, 25.3% had unprotected sex, 16.6% physically injured themselves, and 6.5% seriously considered suicide, associated with their drinking [3].

Several personality domains have been examined for their role in the development and maintenance of alcohol-related problems, with much of the evidence coming from studies with emerging adults [4]. The four-factor model of personality vulnerability identifies four personality-related pathways to increased alcohol use and associated harms [5]: Sensation-seeking (SS), impulsivity (IMP), hopelessness (HOP), and anxiety sensitivity (AS) [5]. SS is the tendency to maximize enjoyable experiences [6]; IMP is acting without planning/deliberation [7]; HOP involves feeling pessimistic about and unable to control the future [8]; AS is fear of anxiety-related physical sensations [8]. Previous studies have shown that these traits are risk factors for alcohol-related problems in student populations, as they increase the probability of adverse alcohol-related outcomes [9]. Specifically, these traits have been shown to predict increased alcohol quantity and frequency [10,11,12] and greater alcohol-related harms [4,13,14,15] among college students.

Protective behavioral strategies (PBS) are behaviors that individuals can engage in to minimize the harms associated with drinking, and thereby decrease the risk for alcohol-related problems [16]. Previous cross-sectional studies and randomized controlled trials have demonstrated that PBS use decreases alcohol use and alcohol-related problems among college samples [17,18,19,20]. This result implies that PBS can be integrated into harm-reduction programs for student populations to potentially reduce alcohol-related harms [17]. Specifically, PBS can potentially be taught to at-risk students [18], such as students who have one or more of the traits in the four-factor model [5].

Previous findings suggest that the reduced use of PBS might at least partially explain why young people with specific at-risk personality traits use more alcohol and experience more harmful consequences from drinking. For example, previous studies revealed that PBS mediate the relationship between impulsive-like traits (e.g., negative urgency) and alcohol use or alcohol-related harms in college students [21,22,23]. This mediation is consistent with results showing that college students high in SS or IMP perceive drinking as a central part of the college culture [24,25], and that reduced PBS mediate the relationship between these beliefs and alcohol outcomes [22]. Even more, this aligns with studies revealing that young people high in SS frequently drink to enhance positive emotions [4,8,26]. Thus, they may avoid engaging in activities that limit the amount of ‘fun’ they could have. This could include avoiding certain PBS, such as failing to avoid drinking games or failing to drink slowly. Similarly, given that the use of PBS requires planning [16], it is less likely that individuals higher in IMP plan PBS in advance and adhere to them.

Previous studies have similarly revealed that reduced PBS use mediates the positive relationships between anxiety and depressive symptoms, and alcohol use or alcohol-related harms in college students (e.g., social anxiety, [27,28]; depressive symptoms, [29]). Previous results have shown that AS is a risk factor for anxiety [6]. Since PBS involve some degree of planning within a social context (e.g., using a designated driver) [16], individuals with higher AS levels may find it difficult to engage in these strategies because they focus their cognitive resources more so on their own anxious thoughts and sensations. Given the associations between AS and avoidant coping tendencies, they also may stay clear of using PBS if they believe that drinking alcohol can help to avoid anxiety symptoms [6]. Previous results have also shown that HOP is a vulnerability factor for depression [6]. Thus, individuals higher in HOP may lack motivation or cognitive resources to plan or perform any PBS and/or may lack concern about alcohol-related harms [6,30]. Further, for all four personality types, the use of PBS in campus settings is complicated by student immersion in a campus cultural context with considerable pressure to use alcohol in risky ways [31].

Despite the evidence suggesting each personality trait in the four-factor model may be associated with decreased PBS use, no research has examined the associations among these four personality traits, PBS use, and alcohol-related outcomes in an overarching mediational model in emerging adults. To the authors’ knowledge, only one study has tested a moderation effect of PBS on the relationship among all four at-risk personality traits and alcohol use among undergraduates, with non-significant results [32]. More research is needed to shed light on the mechanisms underlying the positive associations between traits in the four-factor personality model and outcomes such as alcohol use (alcohol volume), the pattern of use (heavy episodic drinking [HED]), and adverse alcohol-related consequences (harms). We conducted two studies to address these identified gaps, and examined the possible mediating role of reduced PBS use on the known relationships between at-risk personality traits and risky/adverse alcohol outcomes among emerging adults. In our first study, we hypothesized that reduced PBS would partially mediate the relationship between each of the four at-risk personality traits and past 30-day volume of alcohol use in a sample of college students. In our second study, we hypothesized that reduced PBS use would partially mediate the relationships between each trait and both HED and alcohol-related harms in a different sample of college students. By replicating this mediation across two independent samples, our intention was to corroborate and extend previous findings on the relevant role of reduced PBS use in the development of problematic alcohol use among emerging adults. We examined PBS use as a unidimensional construct, based on findings supporting a single higher-order factor of PBS use with college samples [33].

## 2. Study 1

### 2.1. Materials and Methods

#### 2.1.1. Participants

A sample of 922 university students from an Atlantic Canadian university (70.3% females) participated in this study. The mean age was 20.11 years (SD = 2.11). Most participants were completing the first year of their undergraduate degree (33.0%), followed by second (21.1%), fourth (18.7%), third (18.3%), and fifth (5.2%) year; only a small portion were graduate students (3.8%).

#### 2.1.2. Procedure

Research ethics for this study were obtained from St. Francis Xavier University Research Ethics Board (Protocol #23242). All students were invited to complete a 10-min online Survey of Alcohol Use in November of 2017. Participation was voluntary and informed consent was submitted electronically before willing students began the survey. After completing the survey, students were entered in a prize draw for one of three $50 Amazon gift cards as compensation.

#### 2.1.3. Measures

The Substance Use Risk Profile Scale (SURPS) [4] is a 23-item Likert-type scale assessing four personality traits (SS, 6 items; IMP, 5 items; HOP, 7 items; and AS, 5 items). A 4-point Likert scale is used for item ratings from strongly disagree to strongly agree. The SURPS has shown good reliability, construct validity (convergent and discriminant), criterion validity (concurrent and predictive), and content/structural validity [34,35,36]. Internal consistencies were adequate for all subscales in the present study sample (α_SS_ = 0.74, α_IMP_ = 0.75 α_HOP_ = 0.70, α_AS_ = 0.74).

The Protective Behavioral Strategies Scale (PBSS) [18] is a 15-item Likert-type scale that assesses protective strategies such as alternating non-alcoholic with alcoholic beverages, avoiding drinking games, or using a designated driver [18]. Each item is scored on a 5-point Likert-type relative frequency scale, ranging from never (scored as a 1) to always (scored as 5). The PBSS possesses high internal consistency [18,37], construct validity, content validity, and criterion-related validity [18,37,38]. The internal consistency of the total scale was good in the present study sample (α = 0.89).

Participants were asked how many standard drinks containing alcohol they had on a typical day when drinking (quantity). The answer categories were 1–2, 3–4, 5–6, 7–9, or 10 or more, and participants were provided with pictorial examples of standard alcoholic drinks. Students were also asked to indicate how often they had consumed an alcoholic standard drink in the past 30 days (frequency). These quantities/typical drinking day and frequency/past 30-days estimates were multiplied to yield the dependent variable of total volume of past 30-day alcohol use.

#### 2.1.4. Data Analysis

Descriptive statistics for the participant characteristics were generated in SPSS version 25. Bivariate associations were analyzed with Pearson’s R. The direct effect of SS, IMP, HOP, and AS on volume of alcohol controlling for the mediator (c′ paths), path effects of each personality predictor on PBS (a paths), path effects of PBS on volume of alcohol (b paths), indirect effects of SS, IMP, AS, and HOP on volume of alcohol through the PBS mediator (a*b paths), and total effects (c paths) were estimated using maximum likelihood estimation path analysis. Direct effects are the associations between the predictor and the outcome variable unmediated by the hypothesized mediator variable. Indirect effects represent mediated effects. Total effects are the sum of the direct and indirect effects. A single model was computed with the “MODEL INDIRECT” command of Mplus software version 8.3, which uses Sobel’s standard errors [39]. We used the “ANALYSIS” command to request a bootstrap analysis with 1000 bootstrap samples. Standardized path coefficients and corresponding *p*-values are reported.

### 2.2. Results

Among our sample of 922 participants, 30.0% of males reported drinking more than 15 standard drinks/week, and 16.7% of females reported drinking more than 10 standard drinks/week. Table 1 presents means, standard deviations, and Pearson’s r correlation coefficients among the variables.

Table 2 summarizes standardized indirect and direct effects for Study 1 on the left. In the simple mediation path analysis, SS, IMP, HOP, and AS each indirectly influenced past 30-day volume of alcohol consumed through PBS use (a*b). As can be seen in the model (Figure 1), participants with higher SS, IMP, and HOP reported less use of PBS (a paths), which in turn was associated with a greater volume of alcohol use during the past month (b path). On the other hand, participants with higher levels of the AS engaged in more PBS use (a path), which in turn was associated with a lower volume of past month alcohol use (b path). Personality traits accounted for 7% of the variance in PBS use, and PBS use accounted for 28% of the variance alcohol volume.

In terms of direct effects, higher SS and IMP were associated with significantly higher past 30-day volume of alcohol use independently of their association through PBS use (c’; see Figure 1). Alternatively, high HOP and AS were associated with significantly lower past 30-day volume of alcohol use independently of their association through PBS use (c’; see Figure 1).

## 3. Study 2

### 3.1. Materials and Methods

#### 3.1.1. Participants

A sample of 1625 undergraduate students from an Atlantic Canadian university (70.3% females) participated in this study. The mean age was 18.78 years (SD = 1.61).

#### 3.1.2. Procedure

This study involved the use of an archival dataset, from a survey (the Caring Campus Survey) conducted as part of the Caring Campus Project from Fall 2014 to Fall 2016. This Project was a three-year Movember-funded initiative with the overall aim of preventing substance misuse and associated mental health issues on post-secondary campuses [41]. Ethics approval for this study was obtained from the Health Sciences Research Ethics Board at Dalhousie University (REB #2014-3401). A 20-min online survey was sent by email to first-year students. Participation was voluntary, and informed consent was submitted electronically. After completion, students received modest financial compensation ($5 CAD gift card) or partial course credit.

#### 3.1.3. Measures

We used the SURPS [4] to assess the four personality traits (i.e., SS, IMP, HOP, and AS; present sample: α_SS_ = 0.74, α_IMP_ = 0.70 α_HOP_ = 0.88, α_AS_ = 0.72), and the PBSS [18] to assess protective strategies (present sample: α = 0.87).

Item 3 from the Alcohol Use Disorders Identification Test (AUDIT) was used to measure HED frequency [40,42]. This item measures how often the participant had consumed 5 or more drinks (males) or 4 or more drinks (females) on a single drinking occasion during the past term (i.e., gender-specific HED) [43]. Respondents answered on a Likert-type 5-point scale ranging from “never” (scored as 0) to “daily/almost daily” (scored as 4).

The Drinking Harms scale [44] is a 27-item measure of alcohol-related problems. This scale assesses the presence and frequency of alcohol-related problems within the past semester (e.g., passing out or missing class) using a 5-point Likert scale: 0, never, to 4, four or more times a week. The scoring for each item was recoded dichotomously to reduce skew, with a score of 1 indicating the problem was experienced (regardless of its frequency) and 0 indicating it was not. The number of positively endorsed items was counted to determine a total score with a possible range of 0–27 [44]. The Drinking Harms scale (dichotomously scored) has been shown to have good internal consistency (present sample: α = 0.95) [44].

#### 3.1.4. Data Analysis

The same data analytic strategy was employed as described for Study 1. The direct effect of SS, IMP, HOP, and AS on HED and alcohol-related harms controlling for the mediator (c′ paths), path effects of each personality predictor on PBS (a paths), path effects of PBS on HED and alcohol-related harms (b paths), indirect effects of SS, IMP, AS, and HOP on HED and alcohol-related harms through the PBS mediator (a*b paths), and total effects (c paths) were estimated using maximum likelihood estimation path analysis. A single model including both outcomes was computed with the “MODEL INDIRECT” command of Mplus software version 8.3, with a bootstrap procedure to ensure the robustness of the analyses. Standardized path coefficients and corresponding *p*-values are reported.

### 3.2. Results

Among our sample of 1625 participants, 65.7% males and 54.0% females reported at least one HED episode in the past term, and 82.4% males and 85.2% females reported experiencing at least one alcohol-related harm within the past semester. HED and alcohol-related harms were moderately inter-correlated (*r* = 0.56, *p* < 0.01). Table 3 summarizes means, standard deviations, and Pearson’s r correlation coefficients among the study variables.

The mediator model indicated that SS, IMP, HOP, and AS indirectly influenced HED and alcohol-related problems through PBS use (a*b; right side of Table 1). As can be seen in the model (Figure 2), participants with higher SS, IMP, and HOP engaged in less PBS use (a paths), which in turn was associated with more HED episodes and more harmful consequences of drinking during the past term (b paths). In contrast, participants with higher levels of the AS engaged in more PBS use (a path), which in turn was associated with fewer HED episodes and fewer alcohol-related harms (b paths). Personality traits accounted for 8% of the variability in PBS use, and PBS use accounted for 21% of the variability in the alcohol outcomes of HED and alcohol-related harms.

In terms of direct effects, higher levels of SS and IMP were associated with significantly higher HED, and all traits were associated with significantly higher alcohol-related harms, independent of their associations through PBS use (paths c’; Figure 2). Alternatively, higher AS levels were associated with significantly less HED, independently of its association through PBS use, and the direct effect of HOP on HED was non-significant (c’; Figure 2).

## 4. Discussion

This pair of studies examined the effects of PBS as a mediating variable in the relationship between at-risk personality traits and alcohol-related problems (alcohol volume [Study 1], HED, and harms [Study 2]). The indirect (mediated) effects of SS, IMP, and HOP on alcohol volume (Study 1) and HED and harms (Study 2) through reduced PBS use were statistically significant, suggesting a potential mechanism that underlies the association between these personality traits and problematic alcohol use. Additionally, the indirect effects of AS on alcohol-related outcomes through PBS use were significant, but in the opposite direction to our hypothesis. This result suggests that increased use of PBS plays a protective role in higher AS people and that other mechanism(s) must account for the risk pathway of AS with greater alcohol-related harms observed in Study 2. With one exception, all observed mediational results were partial, meaning that the use of PBS only partially explained the associations between the personality traits and the alcohol-related outcomes. In the Study 2, PBS fully mediated the association between HOP and HED. Discussion of the direct and indirect effects, and their implications for research and practice are detailed below.

In accordance with our hypotheses, SS and IMP were significantly and negatively associated with PBS use in both mediation models. This result corroborates previous findings that have revealed reduced PBS use among emerging adults higher in SS-like traits [21,22,23]. Indeed, because those higher in SS are more likely to drink to experience the pleasurable effects of intoxication [4,8,26], they may avoid engaging in activities that limit the amount of ‘fun’ they experience when drinking (e.g., “Staying away from drinking games”). This result could explain why these individuals seem less willing than those lower in SS to engage in PBS such as avoiding drinking games, or leaving the bar at a predetermined time.

The current results also extend previous research [23] in that reduced PBS use partially explained the role of IMP as a risk factor for alcohol-related harms. Impulsive traits characterize many disinhibited disorders that are highly comorbid with alcohol use disorders, such as borderline personality disorder, attention deficit hyperactivity disorder [45], and bipolar disorder [46]. Highly impulsive individuals tend to consume more alcohol and subsequently experience greater alcohol-related harms, which increases the risk of developing alcohol use disorders [23,45,47]. The direct effects of IMP on our alcohol outcomes were also observed in both models; furthermore, out of the four traits examined, IMP was the most strongly positively associated with alcohol volume and harms, followed by SS. Similarly, SS was the most strongly positively associated with HED, followed by IMP. Therefore, our results suggest that providing prevention programs towards emerging adults with disinhibited personality traits may have high value in the prevention of alcohol use disorders. The fact that both effects were partially mediated through decreased PBS use provides a novel target for prevention in personality-matched interventions [48]. This is particularly relevant at the college level, where prevention programs are in place and could be made more nuanced [49].

Consistent with our hypotheses, HOP was significantly and negatively associated with PBS use in both mediation models [32]. Our results complement previous studies that have shown that PBS mediates the relationship between depressive symptoms and problematic drinking [29,30,50]. It is possible that reduced PBS use may reflect the motivational impairments associated with depression and with a vulnerability factor for depression such as hopelessness [51]. Since we did not include a measure of depression, we cannot evaluate if these effects are secondary to the overlap of hopelessness with depression [52]. Theoretically, a chained mediation pathway may exist where HOP increases risk for depression, which results in lesser PBS use, which in turn contributes to alcohol-related harms. Unexpectedly, HOP was directly and negatively associated with alcohol volume, while it showed the expected positive direct association with alcohol-related harms. A possible explanation for this is that high HOP individuals may be less likely than others to attend rewarding social situations (due to their low reward motivation) where drinking takes place, resulting in lesser total volume consumed relative to others. Yet, when they do drink, they drink to cope [4], placing them at risk for experiencing alcohol-related harms.

Contrary to our hypothesis, AS was significantly positively associated with PBS use in both mediation models. Thus, the indirect effect of PBS was protective, in contrast to the direct risk pathway from AS to alcohol-related harms observed in Study 2. Therefore, PBS use does not help explain why AS individuals experience more alcohol-related harms, a risk pathway that has also been supported in previous research [4,5,8,15]. Prior studies have also reported a positive association between AS and PBS use [53,54]. Higher AS individuals may tend to be more cautious due their fear of somatic sensations, so they may use more PBS to avoid the potential adverse physical effects of alcohol [54]. However, this positive association stands in contrast to research showing that decreased PBS use mediates the relationship between anxiety symptoms and alcohol-related problems [55]. These apparently contradictory results may be due to differences between AS and clinical anxiety [32,56]. AS is a trait-like vulnerability associated with fear of physical anxiety sensations. In contrast, clinical anxiety incorporates the experience of a range of emotional/cognitive/somatic symptoms [56] and does not necessarily involve the fear of physical sensations [32]. In the current study, the direct effect of AS on alcohol volume (Study 1) and HED (Study 2) was negative (protective), which may be because higher AS individuals might typically refrain from drinking alcohol excessively. However, when they do drink, higher AS students may experience powerful negative reinforcement, which might contribute to them becoming psychologically dependent on alcohol to cope even if they are not typically drinking in heavy quantities.

Two pathways in the mediational model (HOP with alcohol volume, AS with harms) in the current study imply that a phenomenon known as inconsistent mediation occurred. Inconsistent mediation occurs when the indirect effect and the direct effect have opposite signs, and the direct effect is larger than the total effect [57]. In the case of HOP with alcohol volume, the mediated effect through PBS was a risk pathway, whereas the direct effect was protective. This pattern suggests that there are other mediators in the model that help explain the protective pathway and that the two mediated effects have opposing signs [57]. For example, high HOP individuals may be less likely than others to attend rewarding social situations where drinking takes place, and therefore they might be less exposed to the influence of modeling or social norms [58].

In the case of AS with harms, the mediated effect through PBS was protective and yet the direct effect was a risk pathway. There must be other mediators in the model, with an opposite direction of influence, that explain the risk pathway [57]. Coping and conformity motives [11], as well as generalized anxiety or depression [59], are positively correlated with AS and alcohol-related problems. Thus, they are potential candidates for mediators of the AS to harms risk pathway that works against the protective PBS pathway. AS has been shown to increase one’s risk of developing depression and anxiety, as well as aggravating existing anxiety/depression symptoms [59,60]. The potential mediational risk pathways involved in the relationship between AS and alcohol-related harms might overpower the protective PBS pathway shown in the current study, resulting in overall risk. Indeed, Chinneck et al. [44] showed AS increased risk for alcohol problems through anxiety symptoms and, in turn, coping motives, and that what remained of the AS effect on alcohol problems was protective.

The present study extended the four-factor model of personality vulnerability to alcohol [5], in which specific personality traits serve as risk factors for alcohol-related problems, by indicating a process (decreased PBS use) by which three of these traits (SS, IMP, and HOP) exert their influences as risk factors for alcohol-related problems. In combination with previous literature [18,23,30,37,55], this study supports the use of interventions for targeted at-risk students that incorporate training in PBS [61].

The current results must be interpreted in the context of study limitations. One limitation is that the present samples were approximately 70% female. However, young males tend to experience higher rates of alcohol use and alcohol-related problems [62]. As PBS are more likely to be used by females [53,62], the current results may not generalize to male drinkers. This limitation may also explain the modest-sized correlations observed in the current study, as a sample comprised primarily of females may have lower risk for experiencing alcohol-related problems. Second, the current analyses were cross-sectional. Causal inferences about the relationships among personality traits, PBS use, and alcohol-related harms cannot be based on the results of this study alone. Nonetheless, cross-sectional mediation can be useful when longitudinal data is not available because the indirect effects are still meaningful and of potential practical and clinical significance [63,64]. For example, Linden et al. [55] used cross-sectional data to demonstrate that PBS mediate the relationship between anxiety and alcohol-related problems among college students. Future studies are needed to examine the proposed mediational models longitudinally, to determine if personality is associated with decreased use of PBS over time and whether reduced use of these strategies is associated with escalations in alcohol-related problems over time.

## 5. Conclusions

In summary, the present study extended previous research by showing a mechanism through which SS, IMP, and HOP individuals experience increased alcohol-related problems. People with these personality features have lower use of PBS which increases their risk of experiencing alcohol-related difficulties. Based on this, harm-reduction interventions should incorporate strategies to target the reduced use of PBS among SS, IMP, and HOP college students. Despite a protective pathway from AS to alcohol outcomes via increased PBS use, higher AS students showed higher levels of alcohol-related problems, suggesting the presence of other powerful risk pathways from AS to alcohol-related problems. This finding suggests that PBS are not very relevant as a therapeutic target for higher AS individuals. Other risk pathways, like coping and conformity motives, should be explored and incorporated into harm-reduction interventions for AS individuals.

## Figures and Tables

**Figure 1 ijerph-18-01814-f001:**
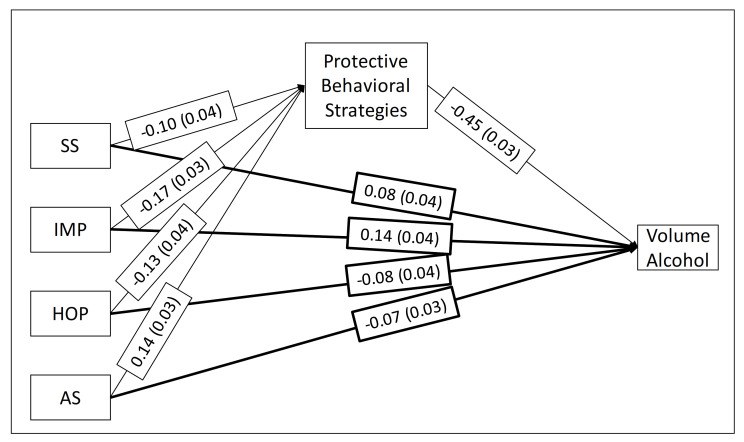
Standardized coefficients for the Study 1 mediational model (a, b, and c’ paths) from Substance Use Risk Profile Scale (SURPS) [4] personality traits to the volume of alcohol used through protective behavioral strategy (PBS) use. SS = SURPS Sensation Seeking; IMP = SURPS Impulsivity; HOP = SURPS Hopelessness; AS = SURPS Anxiety Sensitivity. Light lines represent indirect effects through PBS use, and heavy lines represent direct effects. All paths were significant, *p* < 0.01.

**Figure 2 ijerph-18-01814-f002:**
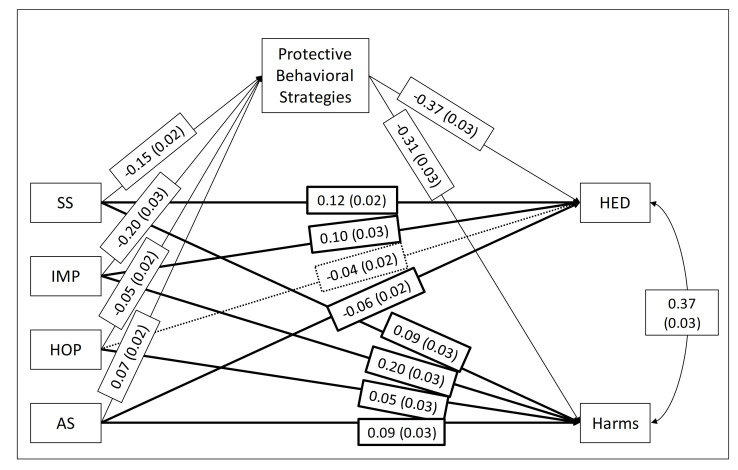
Standardized coefficients for the Study 2 mediational model (a, b, and c’ paths) from Substance Use Risk Profile Scale (SURPS) [4] personality traits to heavy episodic drinking (HED) and alcohol-related harms (Harms) through protective behavioral strategy (PBS) use. SS = SURPS Sensation Seeking; IMP = SURPS Impulsivity; HOP = SURPS Hopelessness; AS = SURPS Anxiety Sensitivity. Light lines represent indirect effects through PBS use and heavy lines represent direct effects. Solid lines represent significant paths; dashed lines indicate non-significant paths, *p* < 0.01.

**Table 1 ijerph-18-01814-t001:** Study 1: Means, standard deviations, and Pearson correlations between personality traits, protective behavioral strategies, and alcohol volume.

Variable	1	2	3	4	5	6	M	SD
1. SS	-	0.29 **	−0.24 **	−0.06	−0.12 **	0.22 **	16.13	3.57
2. IMP		-	0.45	0.20 **	−0.17 **	0.24 **	10.30	2.65
3. HOP			-	0.04	−0.11 **	−0.05	14.58	2.91
4. AS				-	0.11 **	−0.10 **	12.76	2.82
5. PBSS					-	−0.48 **	11.01	2.96
6. Alc. Vol						-	32.94	24.29

SS = Substance Use Risk Profile Scale (SURPS) Sensation Seeking scale; IMP = SURPS Impulsivity scale; HOP = SURPS Hopelessness scale; AS = SURPS Anxiety Sensitivity scale [4]; PBSS = Protective Behavioral Strategies Total Scale [18]; Alc. Vol = alcohol volume (number of standard alcoholic beverages in past 30 days). ** *p* < 0.01.

**Table 2 ijerph-18-01814-t002:** Standardized indirect effects and standard errors of personality traits on alcohol outcomes through protective behavioral strategies, standardized direct effects, and standardized total effects.

	Study 1				Study 2				
	Alcohol Volume			HED			Harms		
	Indirect	Direct	Total	Indirect	Direct	Total	Indirect	Direct	Total
SS	0.04 * (0.02)	0.08 ** (0.04)	0.13 ** (0.04)	0.05 ** (0.01)	0.17 ** (0.02)	0.17 ** (0.02)	0.05 ** (0.01)	0.9 ** (0.03)	0.13 **(0.02)
IMP	0.08 ** (0.02)	0.14 ** (0.04)	0.22 **(0.04)	0.07 ** (0.01)	0.17 **(0.02)	0.17 **(0.02)	0.06 ** (0.01)	0.20 **(0.03)	0.26 **(0.02)
HOP	0.06 ** (0.02)	−0.08 ** (0.04)	−0.02 **(0.03)	0.02 * (0.01)	−0.02 **(0.02)	−0.02 **(0.02)	0.02 * (0.01)	0.05 **(0.03)	0.07 *(0.03)
AS	−0.06 ** (0.02)	−0.07 ** (0.03)	−0.14 **(0.04)	−0.03 ** (0.01)	−0.08 **(0.02)	−0.08 **(0.02)	−0.02 ** (0.01)	0.09 ** (0.03)	0.06 *(0.03)

SS = Substance Use Risk Profile Scale (SURPS) Sensation Seeking; IMP = SURPS Impulsivity; HOP = SURPS Hopelessness; AS = SURPS Anxiety Sensitivity [4]; HED = Heavy Episodic Drinking item 3 from AUDIT [40]. * *p* < 0.05; ** *p* < 0.01.

**Table 3 ijerph-18-01814-t003:** Study 2: Means, standard deviations, and Pearson correlations between personality traits, protective behavioral strategies, heavy episodic drinking (HED), and alcohol-related harms.

Variable	1	2	3	4	5	6	7	M	SD
1. SS	-	0.24 **	−0.17 **	−0.18 **	−0.19 **	0.28 **	0.16 **	16.32	3.55
2. IMP		-	0.30 **	0.15 **	−0.24 **	0.21 **	0.31 **	10.77	2.72
3. HOP			-	0.17 **	−0.08 **	−0.02	0.14 **	13.86	4.02
4. AS				-	0.07 **	−0.11 **	0.07 **	12.74	2.81
5. PBSS					-	−0.42 **	−0.36 **	9.81	2.23
6. HED						-	0.56 **	1.61	1.05
7. Harms							-	5.85	5.52

SS = Substance Use Risk Profile Scale (SURPS) Sensation Seeking; IMP = SURPS Impulsivity; HOP = SURPS Hopelessness; AS = SURPS Anxiety Sensitivity [4]; PBS = Protective Behavioral Strategies Total Scale [18]; HED = Heavy Episodic Drinking item 3 from AUDIT [40]—frequency of HED episodes in past semester; Harms = Drinking Harms Scale [44]—number of alcohol-related harms (problems) experienced in past semester. ** *p* < 0.01.

## Data Availability

The data presented in this study are available on request from the corresponding author. The data are not publicly available as they are archival and ethics approval was not obtained at the time when the data were collected or the informed consent secured that would allow for posting the data in a publicly accessible repository.

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
