# Peer review of "The Mediating Effects of Protective Behavioral Strategies on the Relationship between Addiction-Prone Personality Traits and Alcohol-Related Problems among Emerging Adults"

_ijerph, 2021, doi:10.3390/ijerph18041814_

Round 1
Reviewer 1 Report
Thank you for the opportunity to review the referenced manuscript. The authors did a great job putting this research together.
Although the authors mention borderline personality disorder and ADHD, the reader recommends mentioning the impulsive nature of bipolar disorder.
Combining 2 separate studies into one manuscript is an efficient way of presenting this research. It is however, somewhat difficulty for the reader to follow, making the reader go back to the earlier part of the manuscript at various times to re-read. Recommend improving ease of reading.
Recommend adding implications of the study findings in the summary section.
Author Response
Response to Reviews: Thank you for the helpful comments regarding our manuscript provided by the reviewers. We appreciate the invitation to complete additional revisions and the opportunity to have our paper further considered for publication in the International Journal of Environmental Research and Public Health. Below we address the comments and suggested revisions.
Thank you for the opportunity to review the referenced manuscript. The authors did a great job putting this research together.
Response: We appreciate the review and the positive feedback.
Although the authors mention borderline personality disorder and ADHD, the reader recommends mentioning the impulsive nature of bipolar disorder.
Response: Thank you for this suggestion. We agree with the reviewer on the importance of impulsivity in both bipolar disorder and substance use disorders and the importance of highlighting the comorbidity between both problems. We included borderline personality disorder among the disorders that are highly comorbid with alcohol use disorder. Please see Page 8, line 308.
Combining 2 separate studies into one manuscript is an efficient way of presenting this research. It is however, somewhat difficulty for the reader to follow, making the reader go back to the earlier part of the manuscript at various times to re-read. Recommend improving ease of reading.
Response: Thank you for this valuable observation. We added detail and improved the sections of the paper that direct the reader to previous parts of the manuscript. For example, we improved the Data Analysis section from Study 2, so it is not necessary to go back to study 1 to understand the analysis. We also improved the description of the instruments in Study 2, with the same objective. Please, see Page 6, lines 219-222 and lines 237-245.
Recommend adding implications of the study findings in the summary section.
Response: Following the reviewer’s recommendations, we incorporated the main implications of our study in the summary section. Please see Page 10, lines 403-410.
Reviewer 2 Report
Overall this research article demonstrated PBS as mediating variable between personality traits and alcohol-related issues in a pretty clear way. However the methods part is too general and more efforts could be made for clarification especially for the data analysis section.
Author Response
Response to Reviews: Thank you for the helpful comments regarding our manuscript provided by the reviewers. We appreciate the invitation to complete additional revisions and the opportunity to have our paper further considered for publication in the International Journal of Environmental Research and Public Health. Below we address the comments and suggested revisions.
Overall this research article demonstrated PBS as mediating variable between personality traits and alcohol-related issues in a pretty clear way. However, the methods part is too general and more efforts could be made for clarification especially for the data analysis section.
Response: Thank you for your thoughtful comments. We reviewed the methods section in both studies, adding more detail (please, see Page 6, lines 221-223). Following the reviewer’s recommendations, we also provided more information in the analysis section of both studies. Please, see Page 4, lines 151-163 and Page 6, lines 237-245.
Response to Reviews: Thank you for the helpful comments regarding our manuscript provided by the reviewers. We appreciate the invitation to complete additional revisions and the opportunity to have our paper further considered for publication in the International Journal of Environmental Research and Public Health. Below we address the comments and suggested revisions.
Overall this research article demonstrated PBS as mediating variable between personality traits and alcohol-related issues in a pretty clear way. However, the methods part is too general and more efforts could be made for clarification especially for the data analysis section.
Response: Thank you for your thoughtful comments. We reviewed the methods section in both studies, adding more detail (please, see Page 6, lines 221-223). Following the reviewer’s recommendations, we also provided more information in the analysis section of both studies. Please, see Page 4, lines 151-163 and Page 6, lines 237-245.
Response to Reviews: Thank you for the helpful comments regarding our manuscript provided by the reviewers. We appreciate the invitation to complete additional revisions and the opportunity to have our paper further considered for publication in the International Journal of Environmental Research and Public Health. Below we address the comments and suggested revisions.
Overall this research article demonstrated PBS as mediating variable between personality traits and alcohol-related issues in a pretty clear way. However, the methods part is too general and more efforts could be made for clarification especially for the data analysis section.
Response: Thank you for your thoughtful comments. We reviewed the methods section in both studies, adding more detail (please, see Page 6, lines 221-223). Following the reviewer’s recommendations, we also provided more information in the analysis section of both studies. Please, see Page 4, lines 151-163 and Page 6, lines 237-245.
Response to Reviews: Thank you for the helpful comments regarding our manuscript provided by the reviewers. We appreciate the invitation to complete additional revisions and the opportunity to have our paper further considered for publication in the International Journal of Environmental Research and Public Health. Below we address the comments and suggested revisions.
Overall this research article demonstrated PBS as mediating variable between personality traits and alcohol-related issues in a pretty clear way. However, the methods part is too general and more efforts could be made for clarification especially for the data analysis section.
Response: Thank you for your thoughtful comments. We reviewed the methods section in both studies, adding more detail (please, see Page 6, lines 221-223). Following the reviewer’s recommendations, we also provided more information in the analysis section of both studies. Please, see Page 4, lines 151-163 and Page 6, lines 237-245.
Response to Reviews: Thank you for the helpful comments regarding our manuscript provided by the reviewers. We appreciate the invitation to complete additional revisions and the opportunity to have our paper further considered for publication in the International Journal of Environmental Research and Public Health. Below we address the comments and suggested revisions.
Overall this research article demonstrated PBS as mediating variable between personality traits and alcohol-related issues in a pretty clear way. However, the methods part is too general and more efforts could be made for clarification especially for the data analysis section.
Response: Thank you for your thoughtful comments. We reviewed the methods section in both studies, adding more detail (please, see Page 6, lines 221-223). Following the reviewer’s recommendations, we also provided more information in the analysis section of both studies. Please, see Page 4, lines 151-163 and Page 6, lines 237-245.
Response to Reviews: Thank you for the helpful comments regarding our manuscript provided by the reviewers. We appreciate the invitation to complete additional revisions and the opportunity to have our paper further considered for publication in the International Journal of Environmental Research and Public Health. Below we address the comments and suggested revisions.
Overall this research article demonstrated PBS as mediating variable between personality traits and alcohol-related issues in a pretty clear way. However, the methods part is too general and more efforts could be made for clarification especially for the data analysis section.
Response: Thank you for your thoughtful comments. We reviewed the methods section in both studies, adding more detail (please, see Page 6, lines 221-223). Following the reviewer’s recommendations, we also provided more information in the analysis section of both studies. Please, see Page 4, lines 151-163 and Page 6, lines 237-245.
Response to Reviews: Thank you for the helpful comments regarding our manuscript provided by the reviewers. We appreciate the invitation to complete additional revisions and the opportunity to have our paper further considered for publication in the International Journal of Environmental Research and Public Health. Below we address the comments and suggested revisions.
Overall this research article demonstrated PBS as mediating variable between personality traits and alcohol-related issues in a pretty clear way. However, the methods part is too general and more efforts could be made for clarification especially for the data analysis section.
Response: Thank you for your thoughtful comments. We reviewed the methods section in both studies, adding more detail (please, see Page 6, lines 221-223). Following the reviewer’s recommendations, we also provided more information in the analysis section of both studies. Please, see Page 4, lines 151-163 and Page 6, lines 237-245.
Response to Reviews: Thank you for the helpful comments regarding our manuscript provided by the reviewers. We appreciate the invitation to complete additional revisions and the opportunity to have our paper further considered for publication in the International Journal of Environmental Research and Public Health. Below we address the comments and suggested revisions.
Overall this research article demonstrated PBS as mediating variable between personality traits and alcohol-related issues in a pretty clear way. However, the methods part is too general and more efforts could be made for clarification especially for the data analysis section.
Response: Thank you for your thoughtful comments. We reviewed the methods section in both studies, adding more detail (please, see Page 6, lines 221-223). Following the reviewer’s recommendations, we also provided more information in the analysis section of both studies. Please, see Page 4, lines 151-163 and Page 6, lines 237-245.
Response to Reviews: Thank you for the helpful comments regarding our manuscript provided by the reviewers. We appreciate the invitation to complete additional revisions and the opportunity to have our paper further considered for publication in the International Journal of Environmental Research and Public Health. Below we address the comments and suggested revisions.
Overall this research article demonstrated PBS as mediating variable between personality traits and alcohol-related issues in a pretty clear way. However, the methods part is too general and more efforts could be made for clarification especially for the data analysis section.
Response: Thank you for your thoughtful comments. We reviewed the methods section in both studies, adding more detail (please, see Page 6, lines 221-223). Following the reviewer’s recommendations, we also provided more information in the analysis section of both studies. Please, see Page 4, lines 151-163 and Page 6, lines 237-245.
Response to Reviews: Thank you for the helpful comments regarding our manuscript provided by the reviewers. We appreciate the invitation to complete additional revisions and the opportunity to have our paper further considered for publication in the International Journal of Environmental Research and Public Health. Below we address the comments and suggested revisions.
Overall this research article demonstrated PBS as mediating variable between personality traits and alcohol-related issues in a pretty clear way. However, the methods part is too general and more efforts could be made for clarification especially for the data analysis section.
Response: Thank you for your thoughtful comments. We reviewed the methods section in both studies, adding more detail (please, see Page 6, lines 221-223). Following the reviewer’s recommendations, we also provided more information in the analysis section of both studies. Please, see Page 4, lines 151-163 and Page 6, lines 237-245.

Reviewer 3 Report
Thank you for this very interesting article assessing the mediating effects of PBS on the relationship between personality traits and alcohol related problems.
I found the article very relevant, interesting and well written. I only have very few comments:
Lines 107 – 108: I would suggest that you explain the reader why the studies were performed on two different groups of college students.
Line 121: Study 1 was performed in November 2017. When was study 2 performed? Please specify this under procedure for study 2.
Line 389: How was study 1 funded?
Author Response
Response to Reviews: Thank you for the helpful comments regarding our manuscript provided by the reviewers. We appreciate the invitation to complete additional revisions and the opportunity to have our paper further considered for publication in the International Journal of Environmental Research and Public Health. Below we address the comments and suggested revisions.
Thank you for this very interesting article assessing the mediating effects of PBS on the relationship between personality traits and alcohol related problems.
I found the article very relevant, interesting and well written. I only have very few comments:
Response: Thank you for the positive feedback. We appreciate the reviewers’ recommendations as we believe they helped to improve the paper.
Lines 107 – 108: I would suggest that you explain the reader why the studies were performed on two different groups of college students.
Response: An explanation detailing the importance of replicating a mediational model with PBS across two independent samples has been included in the manuscript. Please, see Page 3, Lines 108-111.
Line 121: Study 1 was performed in November 2017. When was study 2 performed? Please specify this under procedure for study 2.
Response: Thank you for highlighting this lack of consistency across the methods of both studies. We included the date for Study 2 in the procedure. Please, see Page 5, line 210.
Line 389: How was study 1 funded?
Response: Study 1 didn’t receive external funding. This has been incorporated in the Funding section, at the end of the manuscript. Please, see Page 10, line 415.
